# Case Reports from Women Using a Quantitative Hormone Monitor to Track the Perimenopause Transition

**DOI:** 10.3390/medicina59101743

**Published:** 2023-09-28

**Authors:** Maria Meyers, Richard Jerome Fehring, Mary Schneider

**Affiliations:** 1Jefferson County Department of Health, the Children’s Hospital, Birmingham, AL 35226, USA; 2Institute for Natural Family Planning, College of Nursing, Marquette University, Milwaukee, WI 53233, USA; richard.fehring@marquette.edu (R.J.F.); mary.schneider@marquette.edu (M.S.)

**Keywords:** hormonal monitoring devices, natural family planning, fertility, menstrual cycles, perimenopause

## Abstract

The fertility tracking of menstrual cycles during perimenopause with a quantitative hormone monitor is a novel undertaking. Women in regular menstrual cycles have been tracking their fertility using different biomarkers since the 1960′s. Presently, there are newer electronic hormonal devices used to track fertility that provide more exact and objective data to help delineate the fertile time frame of a woman’s cycle. These devices measure quantitative levels of estrogen, the luteinizing hormone, progesterone, and follicle-stimulating hormone, all of which occur at varying levels during the menstrual cycle. As women advance toward menopause, their cycles vary in length, and their hormones fluctuate. In this retrospective analysis, forty-two women aged 40 to 50 tracked their cycles over time, and eight of these forty-two women used the quantitative hormonal device. With the use of this device, the perimenopausal period has revealed distinct hormonal cycle characteristics that are unique to this group of women. It is the purpose of this paper to discuss these cycle’s characteristics during perimenopause, which were found with the use of the quantitative hormonal device.

## 1. Introduction

Perimenopause is a transitional period in a woman’s reproductive life when menstrual cycles become less predictable compared to a woman who experiences regular cycles and who menstruates each month. As women age, they have fewer eggs to ovulate, and there is an increased risk of chromosomal aberrations in the eggs; they have more anovulatory cycles [1], and their hormonal patterns differ compared to younger women in regular cycles [2]. In 2001 and then again in 2010, the Staging of Reproductive Aging Workshop (STRAW) was convened to discuss women’s menstrual cycle characteristics throughout the reproductive continuum [3]. The STRAW criteria can serve as a framework for women to gain knowledge on what reproductive stage they are in, which can help them to anticipate how to track their fertility.

Many women want to track their fertility for self-knowledge and for family planning purposes [4]. They can successfully track their fertility and find their fertile window when their cycles exhibit regular reproductive hormonal characteristics. They can do so with hormonal monitoring devices, cervical mucus, and basal body temperature measurements. However, with perimenopause, defining the beginning and end of the fertile window might be more difficult because of the changing reproductive hormone levels and differing cycle characteristics, as well as an increasing amount of anovulatory cycles [3]. Previous studies in perimenopausal women suggest that women still exhibit evidence of fertility with estrogen and luteinizing hormone surges as markers of fertility [2]. What remains unknown is if these hormonal surges translate to actual ovulatory events.

## 2. Methods

A retrospective descriptive analysis was conducted on the menstrual cycle charts of forty-two women aged 40 to 50 from 21 July 2019 to 21 July 2023. Of these forty-two women, eight of them used a quantitative hormone monitor to track their cycles. This case report concentrates on data that were collected from those eight women and the cycle characteristics with corresponding charts and descriptions of these cycles. These eight women had a history of regular cycles previously during their reproductive life, and none had a history of a medical diagnosis that could rendered their cycles irregular before entering perimenopause.

All women studied received instructions on fertility tracking from a health professional when they signed on for an educational class on the Marquette Method of Natural Family Planning.

Once the eight clients entered care on the first day of instruction, their menstrual cycles were followed every month to 2 months for the duration of this study. During the study, one woman was followed for 4 months of data collection, one woman was followed for 5 months, 2 women were followed for 6 months, one woman was followed for 8 months, one woman was followed for 10 months, one woman was followed for 12 months, and one woman was followed for 17 months. Unfortunately, because this is a retrospective analysis and not an established prospective study; therefore, actual numbers of cycles are not included in this data collection. Because older women often have cycles that can be longer than one month, the number of months does not give the actual number of cycles followed.

All women signed a consent agreement to have their cycles published in a presentation or document upon entering care. No personal identifiers were used in any of the data cycle publications.

To categorize women’s cycle types when they entered care, definitions were established based on the STRAW criteria and other research [2,3]. Below are the definitions used to characterize the cycle types of the forty-two women. It is important to note that all eight of the women using the quantitative monitor entered care in the variable stage of reproduction: the first stage of perimenopause according to STRAW criteria.

Regular cycles: the woman had normal cycle lengths (25–35 days) with normal luteinizing hormone (LH) surges approximately mid-cycle (cycle days 11–16).Postpartum cycle zero: the woman was postpartum and had not yet had menstrual cycles and had not ovulated.Postpartum cycles: the woman was postpartum and having menstrual cycles, but they were longer and had later peak LH surges than a woman in regular cycles.Short cycles: the woman was exhibiting shorter cycles (22–28 days) compared to when she was in regular cycles. This stage also exhibited earlier LH surges (cycle days 9–12). It is important to note that because women’s cycles can vary compared to each other, each woman compared her own previous regular cycles to this new shorter version of her cycle. This period was considered prior to entering perimenopause.Variable cycles: the woman was exhibiting short and long cycles that varied by more than 7 days between cycles. It is notable that STRAW defines this variable stage as cycles differing by 7 days within a 10-month period [3]. This stage could have different characteristics within one woman. For this study, fertility tracking during this time revealed that there were distinct hormonal cycle characteristics common to this variable time that have not been distinctly outlined with the use of a quantitative hormonal monitoring device prior until now.Greater than 60 days without cycles: the woman had a span of 60 days or greater since her last bleeding episode.Menopause: the woman had no cycles in one year.

Women used different methods to track their cycles. When women entered care, they had been using various ovulation and sympto-thermal methods (STM) along with LH stand-alone wands and progesterone stand-alone wands (Proov©, Colorado, USA) to track their fertility. Once they entered care, most of the women (33/42) used the Clearblue© (CB) monitor (Swiss Precision Diagnostics GmbH, Geneva, Switzerland to track their cycles along with LH wands, mucus and/or progesterone wands. Three women used only LH wands and mucus tracking. Eight women used the quantitative hormone monitor, MIRA© (Quanovate Tech Inc., San Francisco, CA, USA) to track their fertility (Table 1). Two women used both the CB monitor and the MIRA monitor at some point of their tracking.

A thorough discussion of cycle-tracking biomarkers is not covered in this paper; however, it is important to discuss that the quantitative hormone monitor, MIRA ©, does allow the accurate quantitative measurement of a woman’s cyclic hormones. This monitor uses immunochromatography with the fluorescence labeling of a woman’s urine assay, and these results connect to an application on the user’s phone through Bluetooth technology [5]. The device measures estrone-3-glucuronide (E3G), a urinary metabolite of estrogen, which is known to rise prior to the luteinizing hormone (LH) during follicular development when women are in regular cycles.

The Clearblue monitor is a qualitative hormone monitor that measures threshold levels of urinary hormones. The advantage of using a hormone monitor that depicts quantitative levels of hormones instead of threshold levels is to depict a more exact fertile window. A comparative study of the CB monitor and the MIRA in women with regular length cycles showed that 60 out of 62 cycles had a rise in quantitative E3G of >100 ng/mL at least 5 days prior to their LH surge [6], signifying adequate knowledge at the beginning of the fertile window. They could then use this information in decisions to avoid or achieve a pregnancy. In some regular cycling women and especially in perimenopause women, however, this rise in E3G might be shorter than 5 days and not give enough forewarning of the impending LH surge. This shortrise could be postulated to be from inadequate follicular development in older women. It is unknown what percentage of these truncated E3G rises prior to the LH surge, resulting in actual ovulation. The MIRA monitor also measures LH, the follicle-stimulating hormone (FSH), as well as pregnanediol glucuronide (PdG), which is a urinary metabolite of progesterone. It is known that LH stimulates ovulation, and progesterone is a marker for post-ovulation confirmation. Preliminary research with MIRA revealed that an adequate cutoff for the beginning of the fertile window might be an E3G level of 100 ng/mL in women who are in regular cycles [6]. In perimenopausal women, there seems to be fluctuations in E3G prior to their LH surge, so a proposed protocol for the beginning of the fertile window occurs when E3G is 100 ng/mL either that day or the day before; otherwise, the woman who has subsequently lower levels of E3G after these high levels can assume infertility at these lower levels. Luteinizing hormone levels of 11 mIU/mL were proposed to signify ovulation, according to Bouchard et al. [6].

## 3. Results

A retrospective analysis of forty-two women aged 40 through 50 who used various means of tracking their fertility for at least 4 months was performed (Table 1).

A subgroup of 8 women from the 42 women used a quantitative urine hormone monitor. This case report seeks to report the daily hormonal fluctuations of these eight women. The quantitative hormone monitor has the advantage over other means of fertility tracking in that it gives accurate urinary levels of hormones throughout a woman’s cycle. These hormone levels can be used to anticipate a woman’s fertile window or allow health professionals to discern if hormone levels are consistent with a probable anovulatory cycle.

Figure 1 shows the forty-two women’s cycle types categorized according to when they entered care. All eight women using the quantitative hormone monitor entered cycles at a variable stage of perimenopause irrespective of age. Their ages ranged from 41 to 50. Although this case report looks specifically at the eight women in variable reproductive states, it is important to note that all 42 women entered care at various stages of reproduction.

The salient point of the cycle characteristics of all forty-two women compared to age at entry of care was that most women of any age could be in any category of cycles. Most postpartum women were younger at the ages of 40–42; however, one woman was postpartum at age 45. Women who were older, aged 47 to 50, were more likely to be in a variable stage of reproduction (the first stage of Perimenopause); however, one woman who was 50 experienced shortened reproductive cycles, which is prior to the variable stage according to the STRAW criteria. There were no women in regular cycles past the age of 47. Of these women aged 40 to 50, 14 entered care in the variable stage, 11 entered care in regular cycles, 9 entered as postpartum cycle zero, 6 entered as short cycles (the last stage of reproductive cycles), and 2 entered care when they had bleeds greater than 60 days between cycles.

While being followed, some women transitioned to another phase of their cycles (13/42). One woman went through menopause early at 44 years old and transitioned quickly from variable stage cycles to >60 days without menses and had no more cycles after that stage, which signified Menopause. Of note, her own mother entered menopause in her 40s as well.

There were no women in regular cycles past the age of 47. This variable stage of reproduction is seen more commonly in women over the age of 46; however, women aged 41, 42, 43, and 45 were already in their variable stage of reproduction at entry to care.

## 4. Discussion

Variable Cycle Stage Characteristics: according to the STRAW study, the variable cycle stage is the first stage of Perimenopause and is when cycles begin to vary by more than 7 days within a 10-month period [3] (Figure 2). However, some cycles in these older women can still be ovulatory within this variable stage. When analyzing the forty-two women’s menstrual cycles, patterns demonstrated some unique hormonal and bleeding characteristics in this period, even within that variability. Fertility tracking during this time revealed that there are distinct hormonal cycle characteristics common to this variable time of reproduction that have not previously been studied with a quantitative hormonal device. This novel information could help us to better characterize women’s cycles. See Table 2.

When looking specifically at the variable stage of perimenopause, women have characteristics earlier that might differ slightly later as they progress through this stage. There can be some overlap between an earlier variable and a late variable stage, as even women in late variable stages can still exhibit regular ovulatory cycles. More exact research is needed to delineate if there is a statistically significant difference in cycles between the early and late variable stages. Some women exhibit normal E3G rises 5 days prior to the LH surge, which can give an adequate fertile window. Some peak LH surges might occur later in the cycle, and those cycles may have shorter luteal phases with longer cycles compared to others in this perimenopausal group of women.

As women progress in the variable stage of perimenopause, they can see truncated E3G rises prior to their LH surge, which might not give an adequate warning of the beginning of the fertile window. They can also see continuous high E3G levels coinciding with high LH levels. It is uncertain if these types of cycles are ovulatory. In what can be assumed to be anovulatory cycles, there can be continuous low levels of both E3G and LH as well. Another characteristic of these variable cycles is double LH peaks, with one peak LH usually greater than a subsequent or previous one, and these LH peaks are usually close together in the cycle. It can be postulated that the greater LH peak is thought to be more likely a true stimulation to ovulation compared to its smaller near-additional LH surge.

There can also be irregular bleeding patterns with short time frames between one bleed and the next bleed with no discernable LH peak. These short episodes might not be ovulatory. PdG levels may be elevated after a true ovulatory cycle; however, in one paper by O’Connor et al., as women age, their PdG levels decline [1]. Declining PdG levels may make it harder to determine if a cycle in this stage is truly ovulatory or anovulatory.

Fertile windows may be better determined with this monitor versus measuring other biomarkers in older women. However, attempting to capture an accurate fertile window in variable cycles is still not without some drawbacks. Sometimes, the E3G rise does not give enough forewarning prior to ovulation. It is unknown what percentage of these truncated E3G rises prior to LH surge results in actual ovulation. There can be cycles that have high E3G and LH continuously, as well as some cycles that contain double LH peaks.

Researching other biomarkers, such as FSH, to find their value when predicting ovulatory versus anovulatory cycles should also be considered. We do know that women with a serum FSH level of 25 or greater drawn three to five days of a cycle are considered to have declining fertility [3]. Additionally, FSH levels are also known to be lower during the peak LH surge in women who are not ovulating [7].

In this following section, there are examples of these types of variable cycles of the eight women out of the forty-two who were tracking with a quantitative monitor. It is important to also note that all charts in these examples were from a variable stage of reproduction. Even in this unpredictable stage, there were some cycles in these women that appeared regular and ovulatory, even in women of older ages. See Figure 3, Figure 4, Figure 5, Figure 6, Figure 7, Figure 8, Figure 9 and Figure 10.

## 5. Conclusions

Tracking fertility during Perimenopause with a quantitative hormonal device is a novel idea. A quantitative hormone monitor allows for the exact measurements of hormones in a woman’s cycles and can give more accurate results of cycle patterns, especially as she ages. This study of women in perimenopause has revealed certain cycle characteristics unique to this period, which include cycles with delayed LH surges, quick rises in E3G toward the LH surge, low E3G and LH levels in a cycle, double LH surges in one cycle with corresponding FSH elevation during the highest LH surge, continuous high levels of E3G and LH throughout the cycle, and low PdG levels after an LH surge.

Future research could use more definitive studies, such as vaginal ultrasounds, to detect which cycles are truly ovulatory versus anovulatory compared with those findings using the quantitative hormone monitor. Continued research in the cycles of perimenopausal women could help women to successfully understand and track their fertility, even as they advance in age.

## Figures and Tables

**Figure 1 medicina-59-01743-f001:**
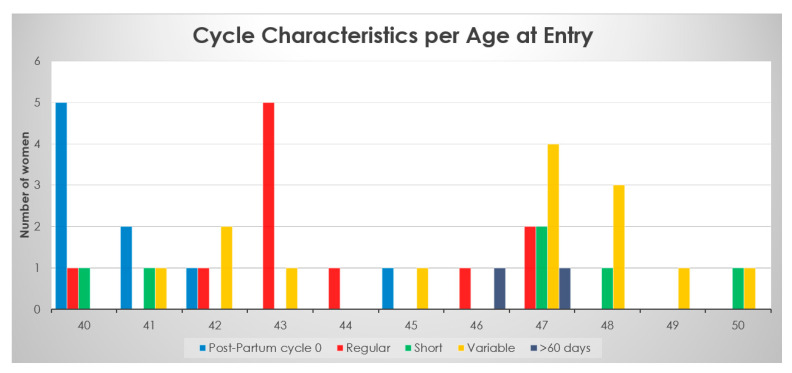
Age, number of women, and their cycles characterized at entry of care (forty-two women).

**Figure 2 medicina-59-01743-f002:**
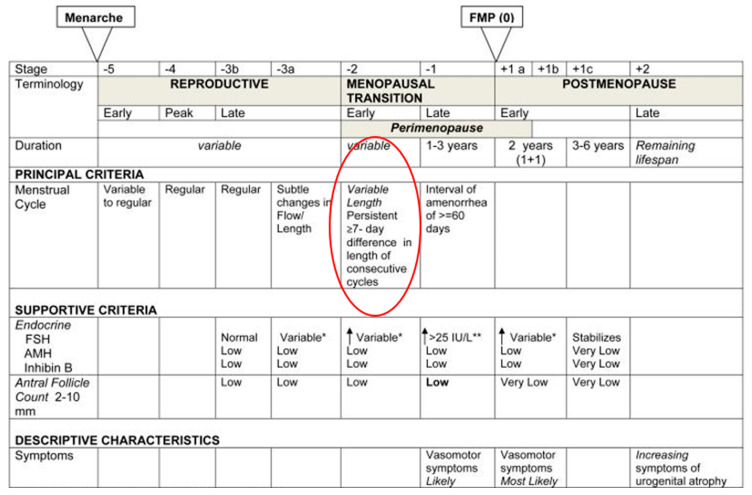
STRAW staging of women’s reproductive cycles. * Blood draw on cycle days 2-5 = elevated; ** Approximate expected level based on assays using current pituitary standard.

**Figure 3 medicina-59-01743-f003:**
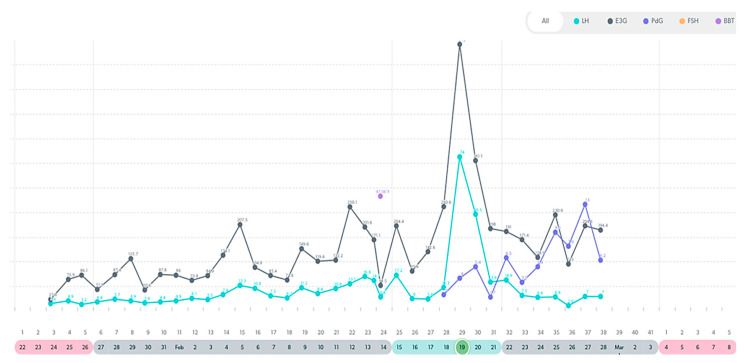
Variable cycle with MIRA: 41 yo-delayed LH peak, 41 day cycle length, PdG rise post ovulation.

**Figure 4 medicina-59-01743-f004:**
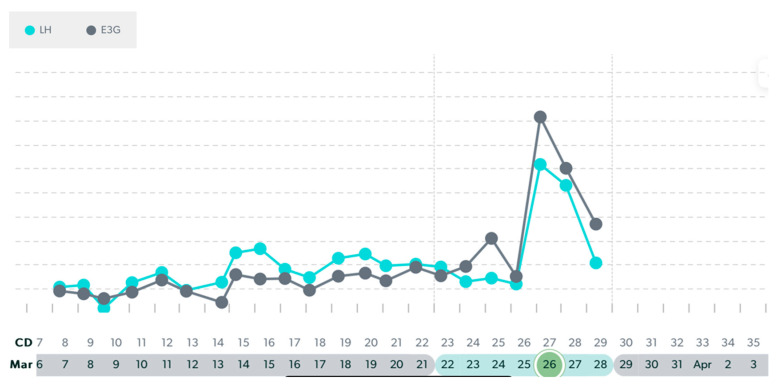
Variable cycle with MIRA: 47 yo with truncated E3G rise to LH peak in a 36-day cycle.

**Figure 5 medicina-59-01743-f005:**
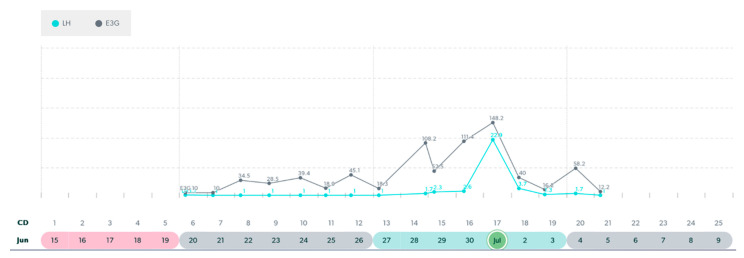
Variable cycle with MIRA: 51 yo with truncated E3G rise prior to LH surge.

**Figure 6 medicina-59-01743-f006:**
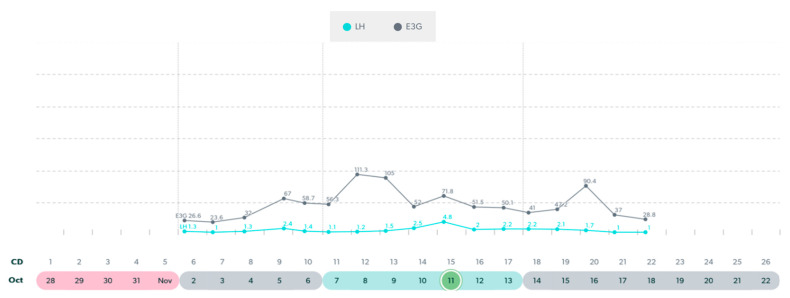
Variable cycle with MIRA: 50 yo with low E3G and low LH: likely anovulatory.

**Figure 7 medicina-59-01743-f007:**
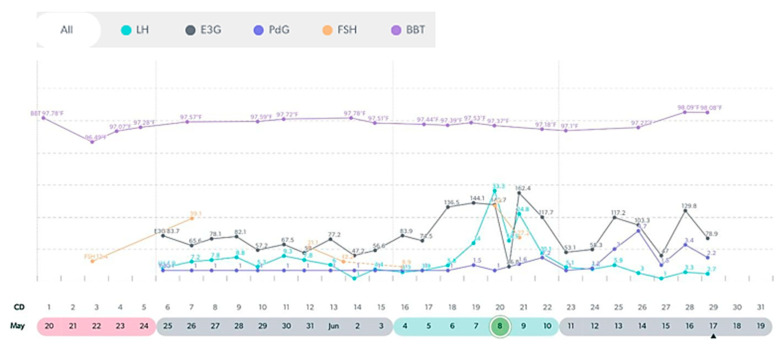
Variable cycle with MIRA: 47 yo with late double LH peak and PdG rise post ovulation.

**Figure 8 medicina-59-01743-f008:**
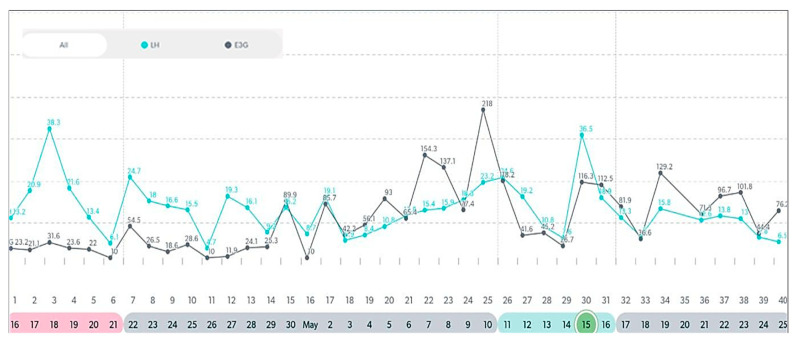
Variable cycle with MIRA: 50 yo with continuous high E3G and LH.

**Figure 9 medicina-59-01743-f009:**
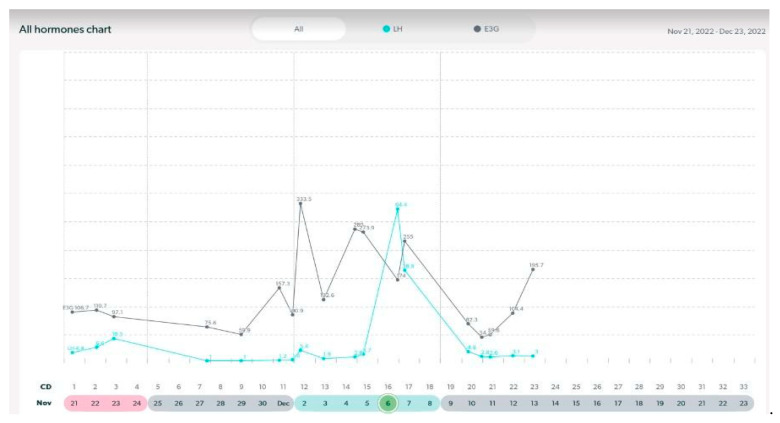
Variable cycle with MIRA: 48 yo with regular cycle; >= 5-day E3G rise prior to LH surge.

**Figure 10 medicina-59-01743-f010:**
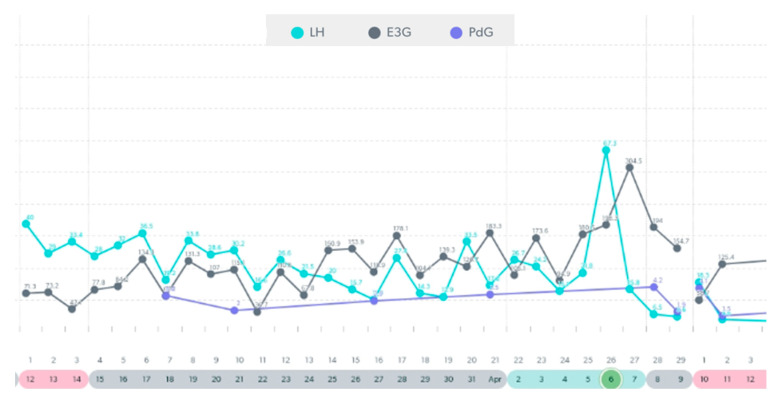
Variable cycle with MIRA: 49 yo with late LH peak, no PdG rise, and a 29-day cycle.

**Table 1 medicina-59-01743-t001:** Methods of fertility tracking used for forty-two women aged 40–50.

Methods Used to Track Fertility	Number of Clients Using That Method	Ages of Clients Using That Method
CB alone or plus mucus, LH wands or PdG wands	30/42	Varying ages btw 40 and 50
LH wands and mucus	3/42	46, 47, 47
STM	1/42	47
MIRA	8/42	41, 46, 47 (3), 49 (2), 50

**Table 2 medicina-59-01743-t002:** Proposed cycle characteristics of pre-perimenopause and perimenopause stages.

	Late Reproductive:	Early Variable:	Late Variable:	Cycles ≥ 60 Days	Menopause: no Cycles for 12 Months
**Cycle length**	-short cycles in the range of 20′s days in length	-variable cycles by 7 days	-variable cycles by 7 days	-months without menstrual flow ≥ 60 days	
**E3G**	-early E3G rise	-E3G rise 6 days prior to LH surge still present	-E3G rise may not be present prior to LH surge or truncated (<6 days)	-low E3G levels	
**LH**	-early LH Peaks, early ovulation	-Peak LH surges occur; some Peaks may be late with short luteal phase	-continues to have Peak level E3G/LH surges without E3G rise	-low LH levels	
**Bleeding** **characteristics**	-bleeding length may be shorter but still regular	-regular bleeding patterns are still present (no “intermenstrual bleeds”)	-irregular bleeding patterns c/w anovulation; long bleeds, short bleeds		
**FSH**					

## Data Availability

The data presented in this study are available on request from the corresponding author.

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
