# Peer review of "Case Reports from Women Using a Quantitative Hormone Monitor to Track the Perimenopause Transition"

_medicina, 2023, doi:10.3390/medicina59101743_

Round 1
Reviewer 1 Report
I would like to congratulate you for the effort you have put into your very fresh and remarkable article. Although your article is very remarkable for the case report category, your article needs to be strengthened.
1-More simplified, removing unnecessary repetitive sentences will be good for readers.
2- Focusing only on the benefits of the new approach will make the article more fluent.
3- It would be very elegant if you give the criticisms of your new approach in the discussion section (e.g. '...the mood questionnaire scores of the participants in the study were not made, which may affect hormone parameters, and since we could not correlate with diurnal rhythm and stress levels, our data are not sufficient to make a definite conclusion ...'.
4-Tables could be more organized and much more elegant.
5- It may be more appropriate to re-evaluate references 5 and 6 as they have the same authors.
I wish you continued success.
The article would look very elegant if it were simplified.
Author Response
I uploaded the latest revision to Case Reports from women using a quantitative hormone monitor to track the perimenopause transition here below, correcting most points provided by both reviewers. Please see atteched file. I also explained my corrections to the reviewers in the comments on the Review Report Form.
Reviewer 1:
1. improved wording to decrease redundancy: 27/28, 59-61, 79-81, 259-260
2 and 3. I added benefits to MIRA on 119, 114-115, added criticisms on 234
4. I cleaned up graphs, enlarged some for improved viewing, added improved wording in table 1
5. I deleted a reference and added one that covered MIRA technology
See uploaded new revision below (full article).

Reviewer 2 Report
I think the article is very clear and focused on the issue. I would like, however, to see if they have any data regarding the role of FSH to help on the monitoring of the cycle for this population since it is a very common marker used for menopause.
Author Response
Thank you for your kind review. I added FSH commentary to the Conclusion section when discussing future research in this area.
Please see revised total document attached.

Round 2
Reviewer 1 Report
I think your article will be ready for publication with minor changes in the relevant sections by adding explanations following the attachment. I wish you continued success and innovative ideas.

Author Response
D1: 50-59: taken out of Intro and put in Discussion as suggested
D2: changed to just "Methods"
D3: Results and Discussion separated. Made Discussion begin with explanation of Variable cycles.
D4-5: changed "Figure 2" as suggested
D 6: removed description of STRAW, however left "Figure 2" labeling as this was referenced in my preceding paragraph
D7: This description is from my observations in this case report and not from anything derived from read literature. So I changed the wording to "some women in this study exhibit...."
D8: This is my own findings and discussed in the Discussion section
D9, D10: I gave reference to O'Connor, et al concerning PdG levels
D11-D19 directions for making all graphs into one single graph is not clear. All graphs depict different hormonal variations of women in perimenopause and it would not be feasible to put all graphs together and then put a summary description of each graph later. This would force the reader to look at the graph and then try to find the description of it at another location.
D20: Moved Conclusion secion to Discussion. Added all types of cycles and their parameters found.
OF NOTE: Figure 8 and 10 were the same, so removed one.
